# Optimal speed in Thoroughbred horse racing

Quentin Mercier, Amandine Aftalion *

Centre d'Analyse et de Mathématique Sociales, CNRS UMR-8557, Ecole des Hautes Études en Sciences Sociales, Paris, France

* amandine.aftalion@ehess.fr

## Abstract

The objective of this work is to provide a mathematical analysis on how a Thoroughbred horse should regulate its speed over the course of a race to optimize performance. Because Thoroughbred horses are not capable of running the whole race at top speed, determining what pace to set and when to unleash the burst of speed is essential. Our model relies on mechanics, energetics (both aerobic and anaerobic) and motor control. It is a system of coupled ordinary differential equations on the velocity, the propulsive force and the anaerobic energy, that leads to an optimal control problem that we solve. In order to identify the parameters meaningful for Thoroughbred horses, we use velocity data on races in Chantilly (France) provided by France Galop, the French governing body of flat horse racing in France. Our numerical simulations of performance optimization then provide the optimal speed along the race, the oxygen uptake evolution in a race, as well as the energy or the propulsive force. It also predicts how the horse has to change its effort and velocity according to the topography (altitude and bending) of the track.

## Introduction

Very little is known about the optimal strategy for a Thoroughbred horse to run and win a race. Because the racing career of a Thoroughbred horse is not so long, and therefore the number of racing opportunities is limited, any information that can help to determine a horse ability according to the race distance or to optimize how to regulate its speed along the race can be crucial.

Due to limitations in the measurement of the mean oxygen uptake ($\dot{V}O2$) for a horse at high exercise, no information is available on the full $\dot{V}O2$ profile in a race, depending on the distance. Up to now, for Thoroughbreds, only measurements on treadmills have been obtained using masks [1–4]. Nevertheless, a portable mask technology has been developed [5], but to our knowledge no track tests have been yet performed on Thoroughbred racehorses. It is only for Standardbred and endurance horses that some study have been made, see [6] for instance. A review book on horse physiology is [7] and some information can also be found in the report [8]. What is known is that horses have a high aerobic capacity, about twice that of human beings, due to a high capacity for oxygen carriage and extractions, as well as a high stroke

**Data Availability Statement:** All relevant data are within the paper.

**Funding:** QM funder: LabEx AMIES (ANR-10-LABX-0002-01) of Université Grenoble Alpes no grant number, program PEPS https://www.agence-maths-entreprises.fr/. The funders had no role in

study design, data collection and analysis, decision
to publish, or preparation of the manuscript.

**Competing interests:** The authors have declared
that no competing interests exist.

volume [9, 10]. They reach the maximal oxygen uptake ($\dot{V}O2_{max}$) much quicker than humans, nevertheless no precise estimate of the time needed to reach a steady state $\dot{V}O2$ is known. Similarly, no information is available about the distance or time at which the $\dot{V}O2$ starts decreasing at the end of a race. Eventually, for long distances, it is not known whether the $\dot{V}O2$ remains almost constant for the most part of the race or oscillates around a mid value, and then at which period and at which amplitude. Estimates on the proportion of energy derived from aerobic and anaerobic pathways during competitive events have been made according to breed and length of race [10] but the relationship that exists between performance and anaerobic capacity remains to be determined. This paper will provide pieces of information for all these issues.

Reference [11] is the only one that we know of where pacing strategy for horses is analyzed, together with the effect of drafting. Several directions of study have been investigated to better understand the effort or mechanical work developed by horses. One of them is to measure the propulsive force either on a force plate [12] or with an instrumented horse shoe [13, 14]. More on the biomechanics of athletic horses can be found in [15] for instance.

The aim of this paper is to provide a mathematical model able to predict how a Thoroughbred horse should regulate its speed over the course of a race in order to optimize performance. We will see how this depends on the distance to run, but also on the shape and topography of the track. It is based on the model developed in [16–18] for human races and it is adapted here to horses to fit the data.

Based on data provided by France Galop and the Mc Lloyd tracking device in French horse races, we are able to model the optimal horse efforts and velocity for a fixed distance, depending on the curvature and change of slopes and ramps. Our model yields in particular information on the $\dot{V}O2$ profile.

## Materials and methods

### Data

The data consist of two dimensional position and speed sampled at 10 Hz for horses racing in Chantilly (France), at the end of the 2019 Thoroughbred horse racing season. The races were all run on a PSF track in a standard surface condition. The data are provided by France Galop the French operational body for flat horse racing, and are from roughly ten races. The tracking system is developed by Mc Lloyd. It is a miniaturized device which does not bring any discomfort to the horse or the jockey. It weighs 90g and is put by the jockey under the saddle. Reliable data is obtained thanks to a patented positioning technology and robust mobile network data transmission, even during crowded events. The accuracy of the system was validated on horses by France Galop by comparing 1ms-accurate photo finish data on more than one hundred races with gap data on the finish line obtained after processing latitude and longitude data: the device mean accuracy was confirmed to be 25cm or 2 hundredth of a second. It is now used by France Galop on all races to provide live position information to the audience. The tracking system provides the latitude and longitude data sampled at 10Hz, as well as the velocity. The latitude and longitude data given by the tracker are projected over a reference track leading to the position of horses in the race which is therefore given as live information to the audience on all races.

### Model

Once we have these raw data, we have to smooth the speed using a third order Savitzky–Golay filter. Therefore, the raw data provide two curves for each horse sampled at 10Hz:

- the curve of time vs distance from start, projected on the reference track, so that each horse runs the same distance,

- the curve of velocity vs distance from start, projected on the reference track.

These curves will allow to determine all the parameters specific to horses.

The model of [16–18] yields an optimal control problem based on a system of coupled ordinary differential equations for the instantaneous velocity $v(s)$, the propulsive force per unit of mass $f(s)$, the anaerobic energy $e(s)$, where $s$ is the distance from start. The system relies on Newton's second law of motion and the energy balance which takes into account the physiology of horses. The energy balance is between the aerobic contribution $\dot{V}O2$ or $\sigma(e)$, the anaerobic contribution $e(s)$ and the power developed by the propulsive force. The mechanical part takes into account the positive slope or negative ramp $\alpha(s)$, the bending of the track and the control on the variations of the propulsive force. A crucial piece of information to be taken care of is the centrifugal force in the bends. For this purpose, the horse is identified with a bending rod. The centrifugal force does not act as such in the equation of movement but limits the propulsive force through a constraint which yields a decrease in the effective propulsive force in the bends.

The physiology of the horse is taken into account through a number of parameters:

- the maximal propulsive force per unit of mass $f_M$,

- the global friction coefficient $\tau$ which encompasses all kinds of friction, both from joint and track. In total, $f_M\tau$ is the maximal velocity,

- the maximal decrease rate and increase rate of the propulsive force which is related to the motor control of the horse: a horse, like a human being, cannot stop or start its effort instantaneously, but needs some time or distance to do it. This is what our control parameters $u_-$ and $u_+$ will provide,

- the total anaerobic energy or maximal accumulated oxygen deficit $e^0$,

- the $\dot{V}O2$ profile as a function of distance, namely the distance $d_1$ at which the maximum of $\dot{V}O2$ is reached, the distance $d_2$ at which $\dot{V}O2$ decreases, and the relative decrease with respect to the maximum value. This is a curve $\sigma(s)$ where $s$ is the relative distance from start, but in fact, in the model, it is a curve $\sigma(e(s))$ where $e(s)$ is the remaining anaerobic energy. The profile of $\sigma$ is to be identified from the data.

These parameters are not measured or given but they are computed numerically for each horse and each race: they are the parameters that allow the optimal control problem to best fit the data, as we will explain below.

Therefore, for a fixed distance, the model predicts the final time, the velocity curve and the effort developed by the horse to produce the optimal strategy. It depends on the geometry of the track, the ramps and slopes and the physiology of the horse.

For the ease of completeness, we provide below the full optimal control problem though the results of the paper do not require to understand it and the reader can skip this paragraph as a first reading. Instead of writing the equations of motion in the time variable, we write them using the distance from start $s$. This amounts to dividing by $v$ the derivatives in time in order to get the derivatives in space. We also write the equations per unit of mass. Let $d$ be the length of the race and $g = 9.81$ the gravity. Let $c(s)$ denote the curvature at distance $s$ from the start, which is provided for each track and $\alpha(s)$ be the slope coefficient. Let $v^0$ be the initial velocity. Let $e^0, f_M, \tau, u_-, u_+$ and the function $\sigma$ be given. They are identified for a specific race and horse. The optimal control problem (where minimizing the final time is equivalent to

minimizing the integral of the inverse of the velocity) coming from the Newton law of motion and the energy conservation is

$$\min_{v,f,e,u} \int_0^d \frac{1}{v(s)} \ ds, \ \ \text{where}$$

$$v'(s) = \frac{1}{v(s)}\left(-\frac{v(s)}{\tau} + f(s) - g\alpha(s)\right), \qquad\qquad v(0) = v^0,$$

$$e'(s) = \frac{\sigma(e(s)) - f(s)v(s)}{v(s)}, \qquad\qquad e(0) = e^0, \ \ e(s) \geq 0, \ \ e(d) = 0,$$

$$f'(s) = \frac{u(s)}{v(s)}, \qquad\qquad u_- \leq u(s) \leq u_+,$$

and under the state constraint

$$f(s)^2 + v(s)^4 \cdot c(s)^2 \leq f_M^2$$

for $s \in [0, d]$.

The optimal control problem for horse performance is solved using Bocop, an open licence software developed by Inria-Saclay France [19]. The discretisation used to solve the system of coupled ODE's is set to one point every two meters which ensure the same accuracy whatever the race. The solution of the optimal control problem yields, for each race and each horse, the optimal effort, regulation of velocity and the $\dot{V}O2$ profile depending on the length and topography of the race. Our aim is therefore to identify the physiological parameters $f_M$, $\tau$, $u_+$, $u_-$, $e^0$, $\sigma(e)$ from the available data.

## Identification process

The identification process is made through a bi-level optimization procedure looking for minimizing errors between the response of a Bocop simulation and the data through the following objective

$$\min_{\mathbf{p}} \left( |t_{simu}(d) - t_{data}(d)| + \frac{1}{N}\sum_{i=1}^N (v_{simu}(s_i) - v_{data}(s_i))^2 \right),$$

where the subscript *simu* (resp. *data*) refers to the variable extracted from the simulation (resp. data). The distance $s_1 = 0$ is the beginning of the race and $s_N = d$ is the length of the race, while $s_i$ are intermediate distances. The parameter $\mathbf{p}$ refers to the vector containing all the variables to identify:

$$\mathbf{p} = (\tau, e^0, f_M, d_1, d_2, \sigma_M, \sigma_f, u_-, u_+),$$

where $\sigma_M$ is the maximal value of $\sigma$ and $\sigma_f$ its final value. The objective is made up of two parts: first the difference in final time at the end of the race $d$, and then the mean square error over the speed measured at $N$ points. For our identification process, $N$ is taken equal to one thousand which ensures a good accuracy for the objective calculation (since it is one point every 2 meters for the longest race as the Bocop calculation) while keeping a relative low time cost. The points are evenly distributed between $s_1$ and $s_N$.

The algorithm used here is a particle swarm optimization method [20] available in the pyswarm library in Python 3, part of the family of the heuristic optimization methods [21]. The main advantage of such a method is its good ability to explore the design space and its ease of use and implementation. A swarm of designs $\{\mathbf{p}_i\}$ is tested, that is the optimal control problem

is solved for these parameters using Bocop, and its objective value is calculated. The performance influences the speed and the direction of the particles inside the design space for the next iteration. At the end of the process, the best score particle is kept as the result. The stopping criterion of the algorithm is set such that the algorithm is unable to find new particles for which the objective is at least $10^{-7}$ better than the best score observed until then. For all the examples treated in this paper, the swarm size is set to 50 particles and a maximum of 150 iterations. Each identification process has reached the stopping criterion described. This method is well suited to our problem since the objective space has a lot of local minimizers so that gradient based methods can get stuck in local minimizers. It insures overall robustness in the results as the design space is always well explored before converging toward a particular area of the design space.

## Topography of the tracks

We have studied three types of races: a 1300 meters, a 1900 meters and a 2100 meters in Chantilly. The GPS track is shown in Fig 1. The 1300m starts with a straight, then there is a bend before the final straight. The 1900m starts earlier with an almost straight and follows the 1300m. The 2100m starts in the final straight of the other races, makes a closed loop before reaching the same final straight and finish line.

The elevation and curvature profiles are provided by France Galop and illustrated in Fig 2. The tracks are made up of straights (zero curvature), arcs of circles (constant curvature) and clothoids (curvature increasing linearly with distance). A clothoid is the usual way to match smoothly a straight and an arc of a circle since the curvature increases linearly. It is used for train tracks and roads as well. It allows smooth variations of velocities which are more comfortable for horses.

The specificity of the track is that there is a bend of 500 meters before the final straight, where the track is going down in the first quarter (about 1.5%) and going up in the second quarter (about 2%). In the 2100 meters, there is also a bend of 400 meters just after the start, which is first going up and then down. We will see that curvature and altitude have a strong effect on the optimal velocity.

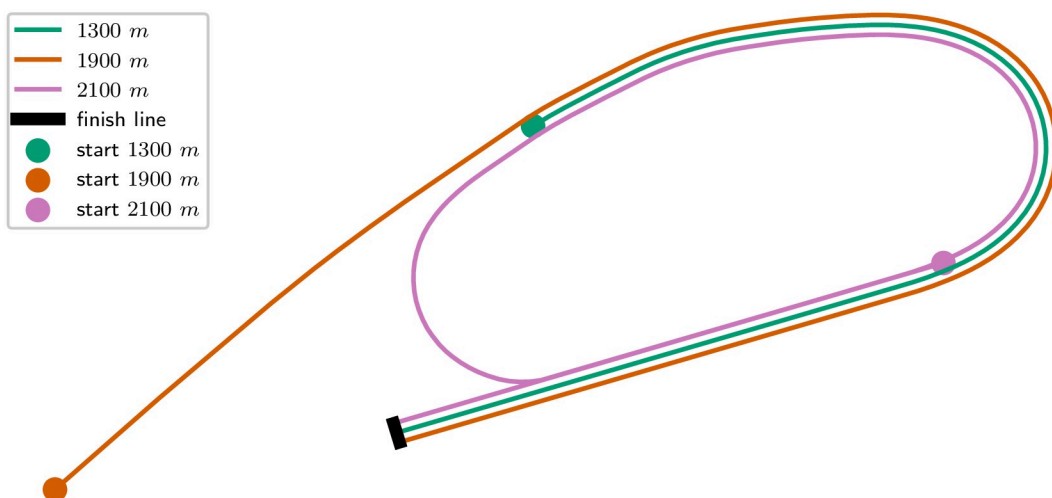

**Fig 1. GPS track of the 1300m, 1900m and 2100m in Chantilly, France.**

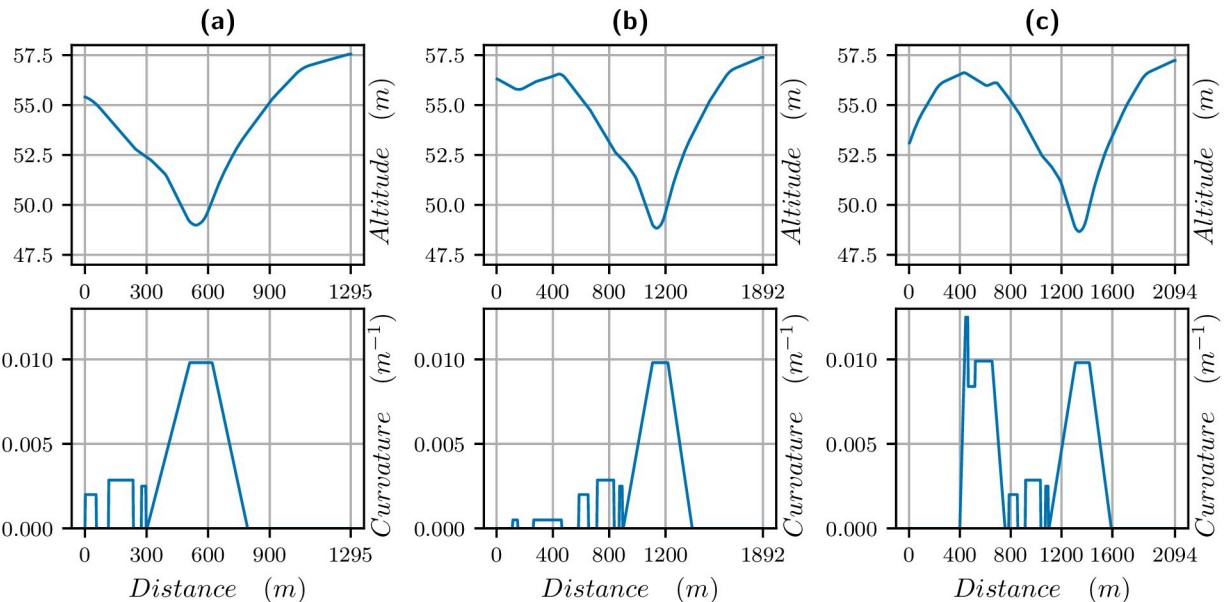

**Fig 2. Altitude and curvature vs distance for different tracks.** (a) 1300m (first column), (b) 1900m (second column) and (c) 2100m (third column). The last 1300m are always the same.

Let us point out that the track is banked but, because data correspond to horses close to the inner part of the track, the banking is not meaningful for the data and will not be taken into account here.

## Results

Among all the available data, we have watched the videos of the races and chosen three races and three horses with the following criteria: no interactions (or at least very little) with other horses that modified the speed, no specific strategy from the jockey to regulate the speed. Therefore, we have chosen horses which seemed to be close to have run a race which would have been similar if they were alone, and could be qualified as their optimal race. We are going to present three significant races of 1300m, 1900m and 2100m. They are all races on a PSF track in a standard surface condition. The 1300m was for 2-year-old horses, the 1900m for 3-year-old and the 2100m for 4-year-old.

We describe below the results of our simulations.

### 1300 meters

The parameters identified for this race are in Table 1. The velocity data (raw and smoothed) and the velocity computed with our model are plotted in Fig 3 for the 1300m. We observe a very good match between the curves: there is a strong start with the maximal velocity being reached in 200 meters. Then the velocity decreases, and in particular in the bend, between 300 and 600 meters from start. Though the track is going down, the centrifugal force reduces the

**Table 1. Identified parameters for the 1300m.**

| $\tau$ | $e^0$ | $f_M$ | $d_1$ | $d_2$ | $\sigma_M$ | $\sigma_f$ | $u_-$ | $u_+$ |
|---|---|---|---|---|---|---|---|---|
| 3.911 | 2731 | 5.150 | 421.5 | 559.9 | 47.0 | 40.71 | -1.693e-03 | 1.504e-03 |

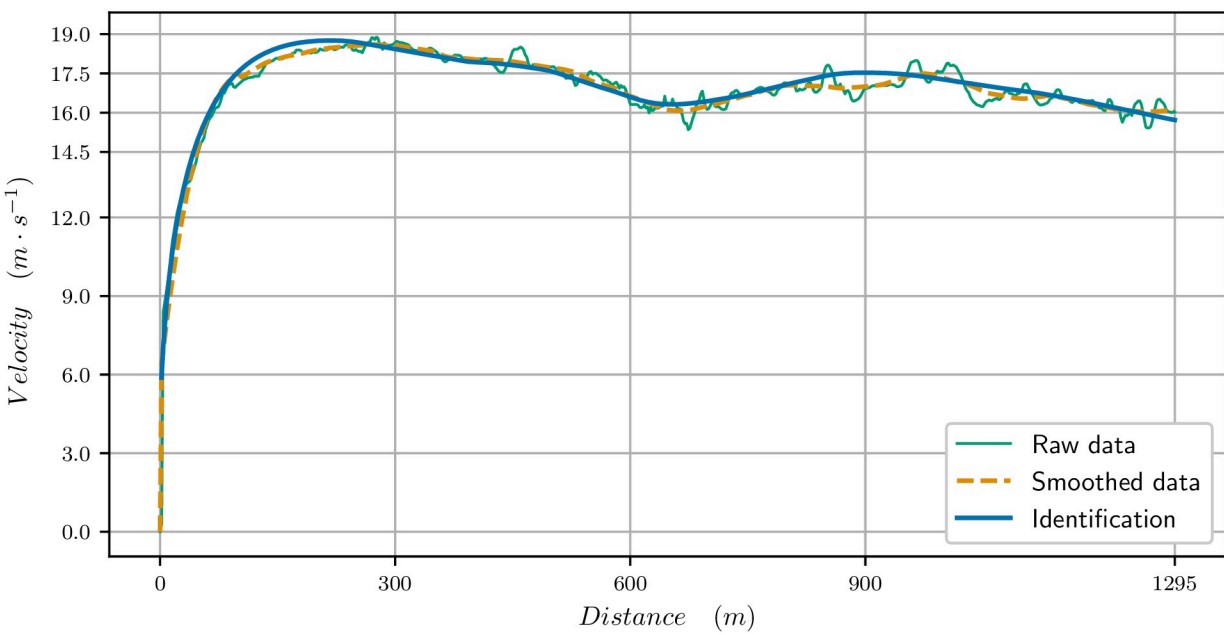

**Fig 3. Velocity data for the 1300 meters race.** Raw data, smoothed data, ($t_f$ = 76.544s) and computed velocity for the identified paramaters ($t_f$ = 76.544s).

propulsive force as we see in Fig 4b (the black curve shows the limitation due to the centrifugal force). It is only when reaching the clothoid, before the final straight, after 600 meters, that the horse can speed up again. The end of the race is uphill and the velocity decreases though the horse reaches the straight. Nevertheless, a decrease in velocity at the end of such a race takes place even on a flat track.

The $\dot{V}O2$ curve vs distance and propulsive force vs distance are plotted in Fig 4. We measure $\sigma$ in $J/s/kg$ but we want to plot the results in terms of $ml/mn/kg$ knowing that one liter of oxygen produces roughly 21 $kJ$ (see [22] and also [23]), so we have to multiply our data for $\sigma$ by 60/21. For a maximal value of $\sigma$ equal to 47, this yields a $\dot{V}O2_{max}$ of 133.6 $ml/mn/kg$. We see that the $\dot{V}O2$ is increasing for about 400 meters, while the force is decreasing. Then when the $\dot{V}O2$ decreases, the force and thus the velocity increase until 900 meters when the slope and

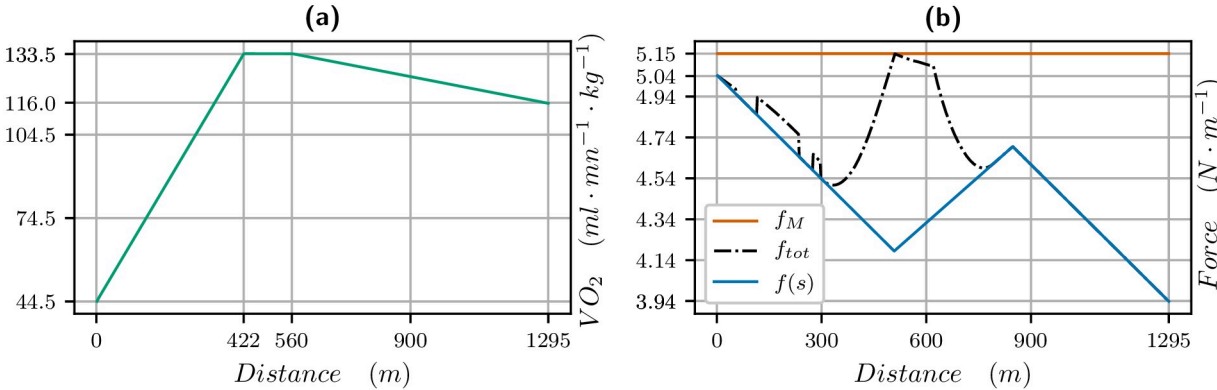

**Fig 4. $\dot{V}O2$ and propulsive force vs distance for the 1300 meters race.** $\dot{V}O2$ (left in green) and propulsive force (right): blue is the propulsive force $f(s)$ in the direction of movement, black is the effective propulsive force $f_{tot} = \sqrt{f^2 + c^2 v^4}$ taking into account the centrifugal force, where $c$ is the curvature.

**Table 2. Identification parameters for the 1900 meters.**

| $\tau$ | $e^0$ | $f_M$ | $d_1$ | $\sigma_M$ | $d_2$ | $\sigma_f$ | $u_-$ | $u_+$ |
|---|---|---|---|---|---|---|---|---|
| 3.73 | 2702 | 4.74 | 379 | 54.6 | 1392 | 40.3 | -3.69e-03 | 3.94e-03 |

end of race lead to a decrease of force and velocity. We point out that the value of the propulsive force is higher than the ones found in [13, 14] but the velocity is also much higher.

## 1900 meters

The parameters identified for this race are in Table 2. The velocity data (raw and smoothed) and the velocity computed with the model are plotted in Fig 5 for the 1900 meters. We observe that there is a strong start with the maximal velocity being reached in 300 meters. Then the velocity decreases. Between 900 and 1400m, we see the effect of the bend: at the beginning of the bend, the track is going down and the horse slightly speeds up; then the centrifugal force reduces the velocity but the velocity increases again at the end of the bend. The end of the race is with a strong acceleration before the final slight slow down. The $\dot{V}O2$ curve vs distance and propulsive force vs distance are plotted in Fig 6. We see that $\dot{V}O2$ is increasing for about 400 meters, while the force starts at maximal value. Then, the $\dot{V}O2$ is constant, the force and the velocity decrease to a mean value. At the end, the $\dot{V}O2$ decreases when the residual anaerobic energy reaches a third of its initial value. The effect of the bend and centrifugal force are obvious: it leads to a decrease in propulsive force and velocity. We see on the force profile that there is a very strong acceleration in the end. It can only take place after the bend where the centrifugal force reduces the available propulsive force.

In Fig 7, we have plotted a zoom on the velocity curve for the identified parameters, and then have removed the effect of the slope (flat track), of the curvature (straight track) and of

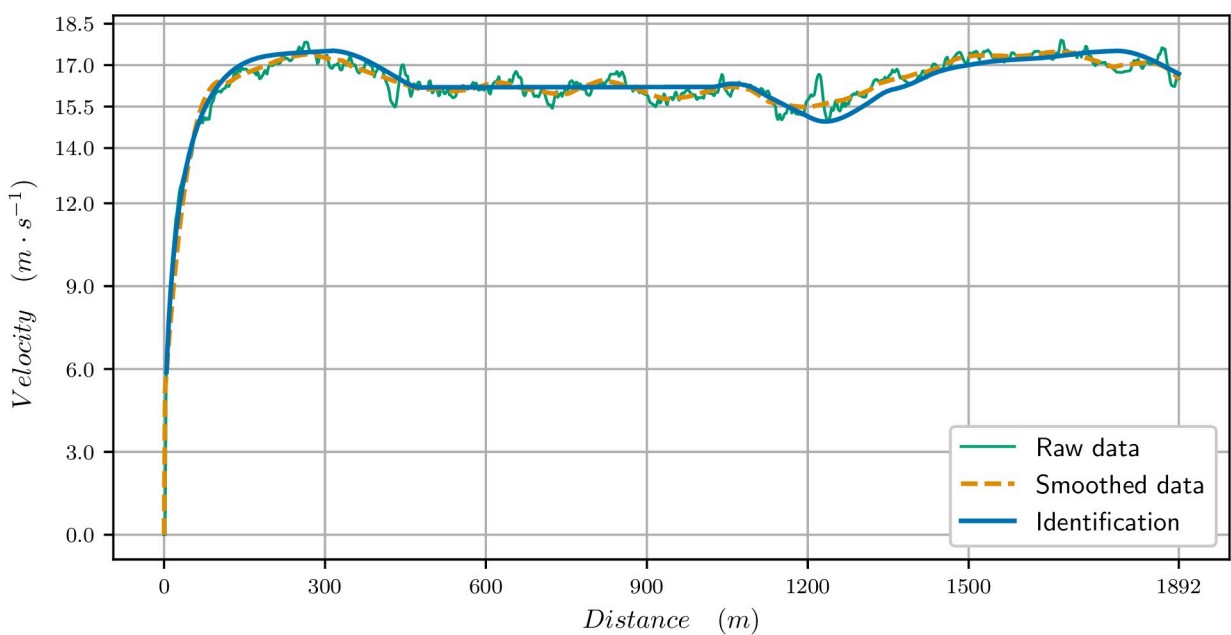

**Fig 5. Velocity data for the 1900 meters race.** Raw and smoothed data ($t_f = 116.460$s) and computed velocity ($t_f = 116.460$s) for the identified parameters.

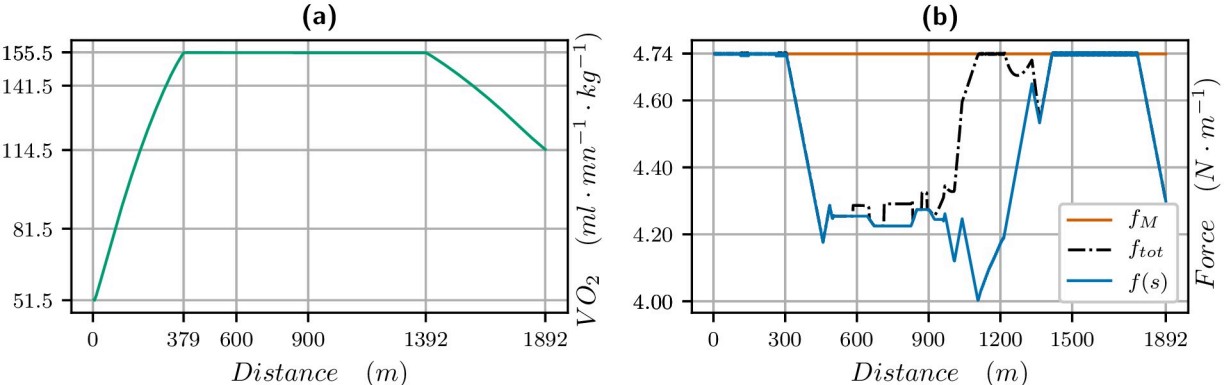

**Fig 6. $\dot{V}O2$ and propulsive force vs distance for the 1900 meters race.** $\dot{V}O2$ (left in green) and propulsive force (right): blue is the propulsive force $f(s)$ in the direction of movement, black is the effective propulsive force $f_{tot} = \sqrt{f^2 + c^2 v^4}$ taking into account the centrifugal force, where $c$ is the curvature.

both (flat, straight track). This allows to notice the specific effects: a bend reduces strongly the velocity (pink curve); on the real track (brown curve), because the first part of the bend is going down, the reduction in propulsive force and velocity is not so strong. The end of the track is uphill and one notices that the velocity curves corresponding to going up (brown, red) cannot provide a speeding up as high as the two others. The combination of slopes and ramps of this track (red curve) reduce the velocity and final time in total though the profile is very similar. We also point out that though these are local effects, they have a global influence on the strategy since they change the mean velocity.

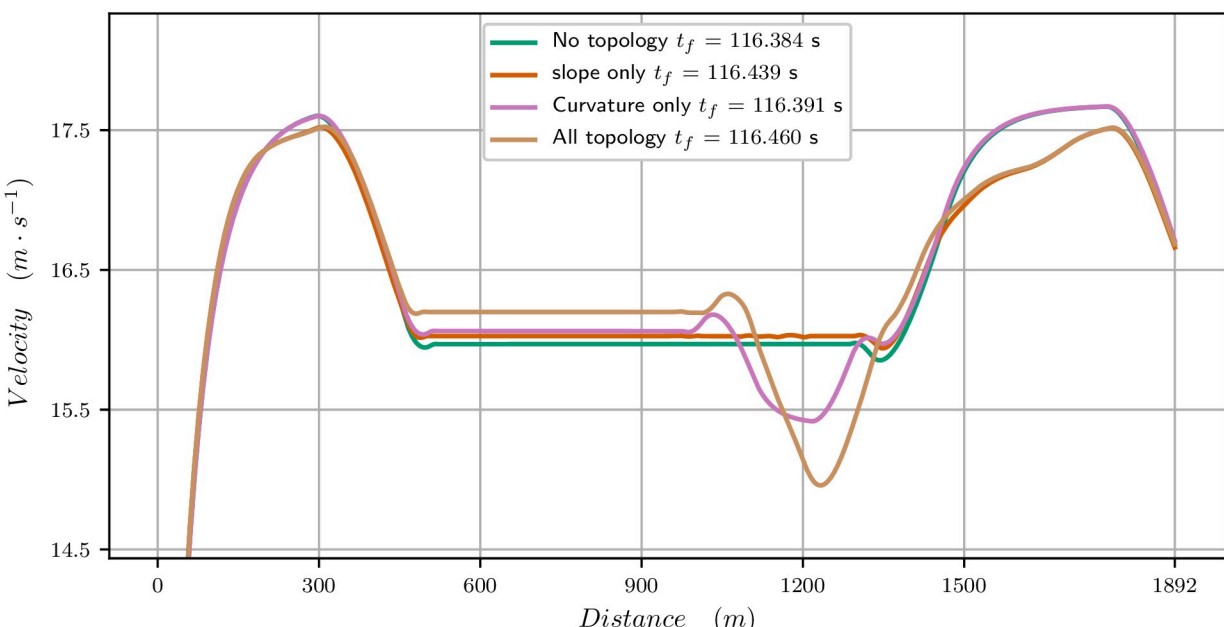

**Fig 7. Effect of the slope and curvature on the velocity curve (zoom) for the 1900 meters race.** Brown is the velocity curve of the race, red with the slope only (straight track), pink with curvature only (flat track) and green is a flat, straight track.

**Table 3. Identification parameters for the 2100 meters.**

| $\tau$ | $e^0$ | $f_M$ | $d_1$ | $\sigma_1$ | $d_2$ | $\sigma_M$ | $d_3$ | $\sigma_f$ | $u_-$ | $u_+$ |
|---|---|---|---|---|---|---|---|---|---|---|
| 3.637 | 3567 | 5.50 | 304 | 51.29 | 524 | 46.71 | 1613 | 41.67 | -1.928e-03 | 9.922e-04 |

### 2100 meters

The parameters identified for this race are in Table 3. The velocity data and the computed velocity are plotted in Fig 8 for the 2100 meters. The $\dot{V}O2$ and propulsive force are plotted in Fig 9. Here, the $\dot{V}O2$ is modified to match the behaviour observed in [24] for humans where the $\dot{V}O2$ first reaches a peak value, before the mean race value. The first bend going up requires a rise of $\dot{V}O2$ at the beginning of the race. We observe that there is a strong start with the maximal velocity being reached in 200 meters. Then the velocity decreases and reaches a plateau. This plateau has been analyzed for human race in [25] and is related to a turnpike phenomenon. It is very likely that the horse optimal velocity for long races can be analyzed with this mathematical tool as well.

The first bend has a strong curvature and therefore reduces drastically the velocity as we can see in Fig 9b: the propulsive force is reduced in the first bend. In the last bend, as in the previous race, the velocity decreases and increases again at the end of the bend. The end of the race is similar to the 1300 meters, with a strong acceleration before the final slow down. The horse in this race is not as good in terms of performance as the one in the 1900m and he cannot maintain his velocity similarly at the end of the race.

## Discussion

### Results on $\dot{V}O2$

From experiments on human races from 400m to 1500m (that is of similar duration of the races we analyze here) [24, 26], it is expected that the $\dot{V}O2$ curve vs distance is

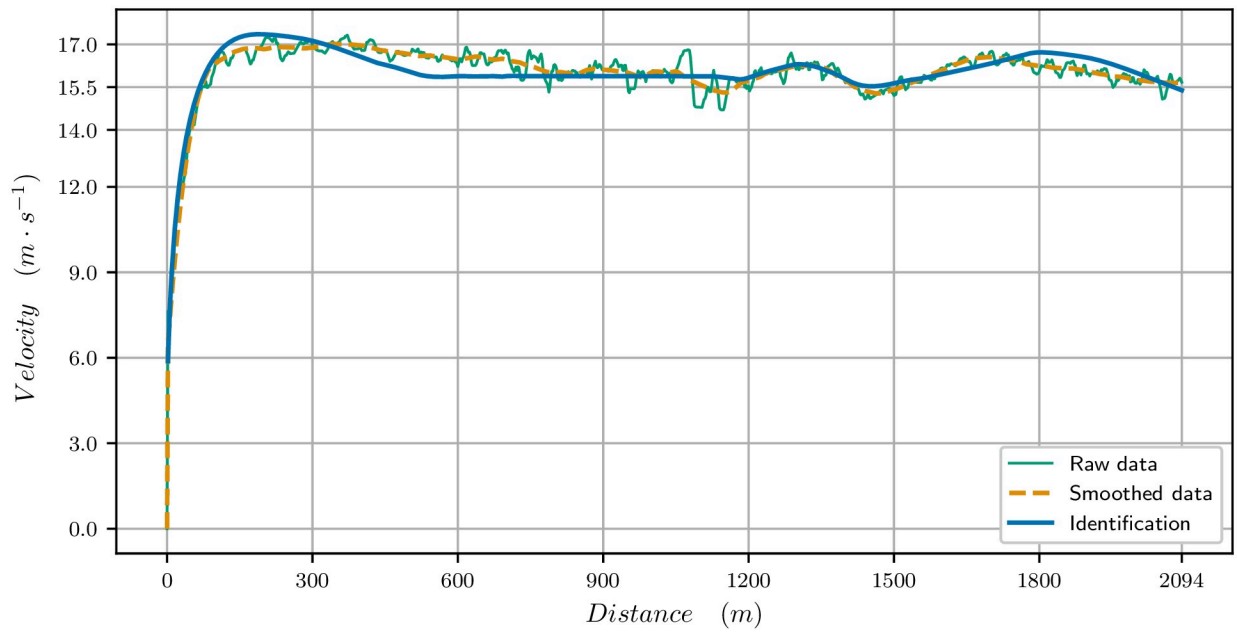

**Fig 8. Velocity data for the 2100 meters race.** Raw and smoothed data $t_f$ = 130.933s) and computed velocity ($t_f$ = 130.933s) for the identified parameters.

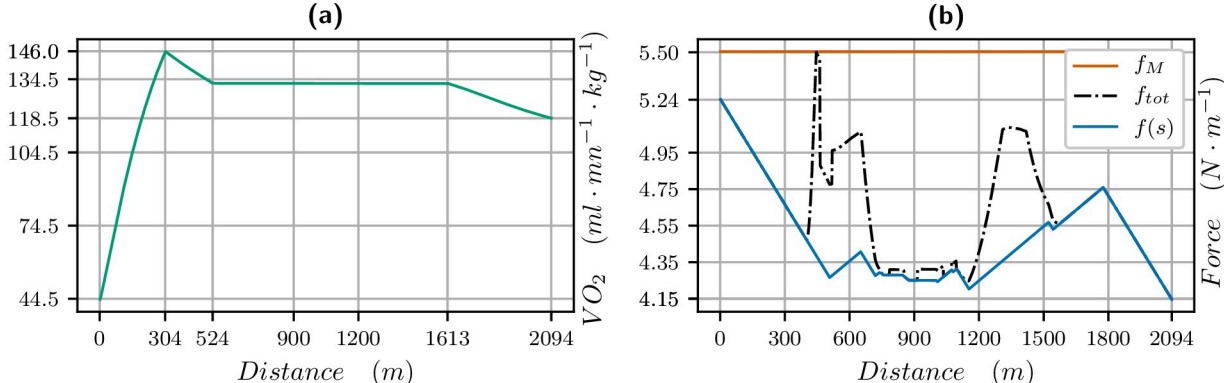

**Fig 9. $\dot{V}O2$ and propulsive force vs distance for the 2100 meters race.** $\dot{V}O2$ (left in green) and propulsive force (right): blue is the propulsive force $f(s)$ in the direction of movement, black is the effective propulsive force $f_{tot} = \sqrt{f^2 + c^2 v^4}$ taking into account the centrifugal force, where $c$ is the curvature.

- increasing to a maximal value and then decreasing for short exercises,

- increasing and reaching the maximal value $\dot{V}O2_{max}$, and then decreasing at the end of the race when the residual anaerobic energy is less than 30%,

- reaching a peak value which is higher than the value along the race for moderate length exercises.

Our simulations and identifications yields that the behaviour is the same for horses. The results of our simulations even provide precise information on the $\dot{V}O2$ curve all along the race. We see in Figs 4a, 6a and 9a, that

- the maximal value of $\dot{V}O2$ is reached in about 400m, that is about 20 to 30 seconds from start, which is indeed much quicker than humans, (this is consistent with [1]),

- the 1300 meters is a short exercise where $\dot{V}O2$ increases and decreases,

- for the 1900 and 2100 meters race, the $\dot{V}O2$ reaches a mean value during the race and decreases about 500 meters from the finish line, when, as for humans, the residual anaerobic energy is about a third of the initial value. The launch of the sprint is optimal when the $\dot{V}O2$ starts decreasing at the end of the race, and this follows from optimal control theory. Numerically, we can observe that the decrease in $\dot{V}O2$ and increase in velocity at the end of the race take place at the same time,

- in a 2100 meters race, the $\dot{V}O2$ first reaches a peak value before decreasing to a plateau value.

The longer the horse can maintain its maximal value $\dot{V}O2_{max}$, the better the performance is. Because the change of slope in $\dot{V}O2$ is related not exactly to the distance but to the available stock in anaerobic capacity, a high anaerobic capacity is all the more important to maintain high velocities all along the race. A strong acceleration at the beginning of the race allows to reach the maximal value of $\dot{V}O2$ quickly and is the best strategy. Jockeys often start slower than the optimal strategy being afraid that if the horse accelerates too strongly with respect to its capacity at the beginning of the race, then the drop in velocity at the end of the race will be bigger.

It is very likely that for horses, as for humans and explained in [24], results on treadmills are very different from measurements during a race. Indeed, as pointed out in [24], exercise at constant velocity yield contradictory results with exercises in a real race in terms of $\dot{V}O2$ or pacing strategy. On a treadmill, the $\dot{V}O2$ increases to reach a maximum value, whereas on a real race a decrease of $\dot{V}O2$ is observed. Our numerical simulations obtained with varying parameters around the identified ones illustrate that in many cases, the best performance is achieved with a fast start, when the pace at the beginning of the race is higher than the pace at the end. While a fast start helps to speed up $\dot{V}O2$ kinetics, and limits the participation of the anaerobic system in the intermediate part of the race, nevertheless, if it is too fast, it has the potential to cause fatigue and have an overall negative effect on the performance. Therefore, a departure which is too fast with respect to the horse's capacity increases the participation of the anaerobic system at the beginning of the race and can damage the final performance. Eventually, a fast start does not necessarily induce a good performance. But a high value of $v$ $\dot{V}O2_{max}$ can allow a faster start velocity without increasing the $O_2$ deficit.

As soon as there is a slope or ramp, the $\dot{V}O2$ is also impacted. Here, in Chantilly, the slope coefficients are strong enough to change the optimal velocity but not strong enough to modify the overall $\dot{V}O2$ profile. The only effect is on the optimization of performace. For stronger slopes, one would need to take into account additionally that $\dot{V}O2$ increases with slopes both downhill and uphill as explained in [27] and [28].

## Energy

Horses have two distinct types of energy supply, aerobic and anaerobic. In our model, it is $e^0$ which estimates the anaerobic energy supply. For human races, recent research suggest that energy is derived from each of the energy-producing pathways during almost all exercise activities [29], which is what we also observe in our simulations.

In Table 4, we have computed the percentage of anaerobic energy to the total energy according to the length and duration of the race. The horse of the 1900m race has a very strong $\dot{V}O2_{max}$ and therefore uses a lower anaerobic energy. We point out that the values estimated in [1] on a treadmill seem to be under estimated with respect to ours: for an exercise of duration 130 seconds, they find an anaerobic contribution of around 30%, which is smaller than our value. Indeed, in a race, for a similar duration of exercise, velocities and forces are much higher than on a treadmill, leading to a bigger contribution of the anaerobic supply [29].

Fig 10 provides the evolution of the anaerobic energy of the three horses vs the distance to the finish line. We have highlighted the points where the $\dot{V}O2$ changes slope. There is a strong anaerobic energy consumption both at the beginning and end of the race, that we can identify through strong slopes. The beginning of the race corresponds to the distance to reach $\dot{V}O2_{max}$.

The horse of the 1300m is a young horse and he needs more time (or a bigger distance) to reach $\dot{V}O2_{max}$ (1214m vs 839 or 889 for the two others). The horse of the 1900m has a higher $\dot{V}O2_{max}$, therefore he needs less anaerobic energy when he is at $\dot{V}O2_{max}$ to maintain his speed. Indeed, once the maximal oxygen uptake is attained, the extra energy to increase speed further

**Table 4. Percentage of anaerobic contribution in the total energy during the race.**

| Race length ($m$) | duration ($s$) | $\dot{V}O2$ deficit |
|---|---|---|
| 1300 | 76 | 47.32% |
| 1900 | 116 | 32.09% |
| 2100 | 131 | 38.01% |

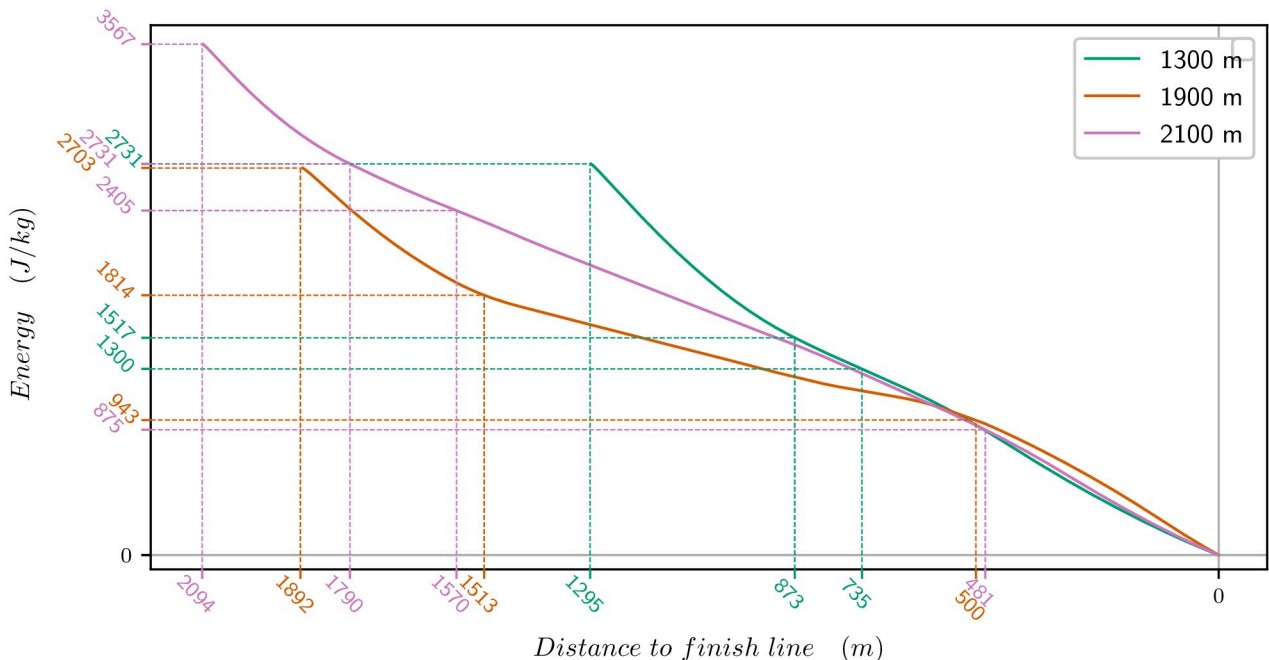

**Fig 10. Evolution of the anaerobic energy vs the distance to the finish line.** Green: horse of the 1300m, orange: horse of the 1900m, pink: horse of the 2100m. The distances are highlighted when the energy has a change of slope and the energy value at this point is indicated as well: it is at the beginning of the race when the $\dot{V}O2_{max}$ is reached, and at the end of the race, when the $\dot{V}O2$ starts decreasing.

is provided by anaerobic pathways. So if a horse has a higher $\dot{V}O2_{max}$, he will use less anaerobic energy to run at the same speed as a horse with a lower $\dot{V}O2_{max}$. In total, this explains why the horse of the 1900m has a lower contribution of the anaerobic energy to the total energy.

### Effect of slopes, ramps and bends on a race

As evident from the data of [30], on a tight bend, horses slow down a lot. In Chantilly, the bends have a radius of at most 100m, which is not tight, but still has a strong impact on the optimal velocity.

To better illustrate this effect, we choose a race of 1900 meters and set an imaginary slope or ramp or bend for one third of the race at the beginning, middle or end of the race (that is roughly 630m) with the following configuration:

- either a positive slope of +3% for 630m,

- or a negative slope of −3% for 630m,

- or an arc of circle with a curvature of $1/100m^{-1}$ for 630m.

Figs 11, 12 and 13 provide the optimal velocity vs distance for the 1900 meters parameters. The common feature is that a local change of elevation or bend does not only produce a local change in velocity but changes the whole velocity profile and mean value.

For a slope going up, as illustrated in Fig 11, the best time is obtained when the slope is at the end of the race. Indeed, a good horse can still provide a strong effort at the end of the race, even if he is tired. If the slope is at the beginning, it has a strong effect on the velocity which cannot reach its maximum value. In the middle of the race, the slope reduces the mean velocity and therefore the final time.

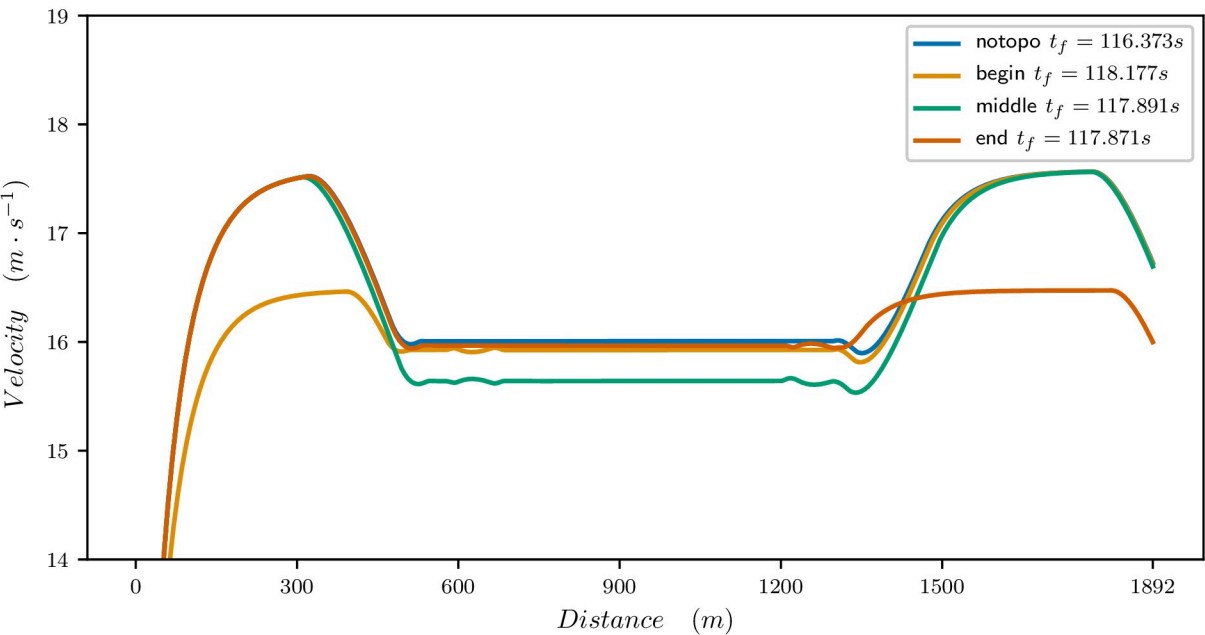

**Fig 11. Effect of slope on the velocity profile for a 1900m race.** Flat straight track (blue), +3% slope for 630m at the beginning of the race (orange), +3% slope for 630m at the middle of the race (green), +3% slope for 630m at the end of the race (red).

For a ramp going down, as illustrated in Fig 12, it is the opposite: the best time is obtained when the ramp is at the beginning of the race. Indeed the horse speeds up more easily and more quickly.

For the bend, as illustrated in Fig 13, the best time is obtained when the bend is at the middle of the race. This is where it has the smallest decrease on the velocity profile. At the beginning, it prevents the horse from reaching its maximal speed. Of course, here the effect is exaggerated with respect to a real race because the bend is long but it yields the general flavour. Similarly, at the end, it prevents the horse from sprinting.

Therefore, we have seen that the optimal velocity has to be analyzed with respect to the changes of slopes, ramps or bends in order to optimize the horse effort. For a given track, it is very likely that the turnpike theory of [25] should yield more precise and detailed analytical estimates of the increase of decrease of velocity.

## Conclusion

Thanks to precise velocity data obtained on different races, we are able to set a mathematical model which provides information on how horses have to regulate their speed and effort on a given distance. It relies on both mechanical, energetic considerations and motor control. The process consists in identifying the physiological parameters of the horse from the data. Then the optimal control problem provides information on the velocity, the propulsive force, the $\dot{V}O2$ and the anaerobic energy. We see that horses have to start strongly and reach a maximal velocity. The velocity decreases in the bends; when going out of the bend, the horse can speed up again and our model can quantify exactly how and when. The horse that slows down the least at the end of the race is the one that wins the race. We understand from the optimal control problem that this slow down is related to the anaerobic supply, the $\dot{V}O2_{max}$ and the ability to maintain maximal force at the end of the race. Therefore, horses that have a tendency to slow down too much at the end of the race should put less force at

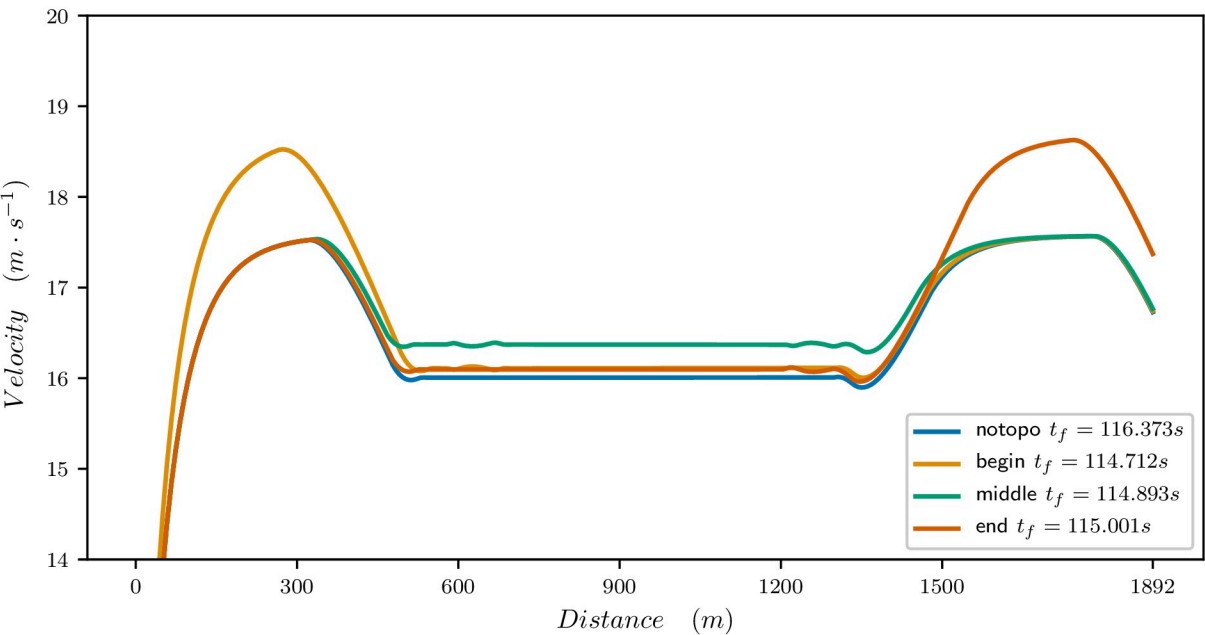

**Fig 12. Effect of ramp on the velocity profile for a 1900m race.** Flat straight track (blue), +3% ramp down for 630m at the beginning of the race (orange), +3% ramp for 630m at the middle of the race (green), +3% slope for 630m at the end of the race (red).

the beginning and slow down slightly through the whole race in order to have the ability to maintain velocity at the end.

From our simulations, we are also able to get information on the $\dot{V}O2$ profile, such as when steady state $\dot{V}O2$ is reached, when the decay of $\dot{V}O2$ starts. The ability to maintain a high $\dot{V}O2$

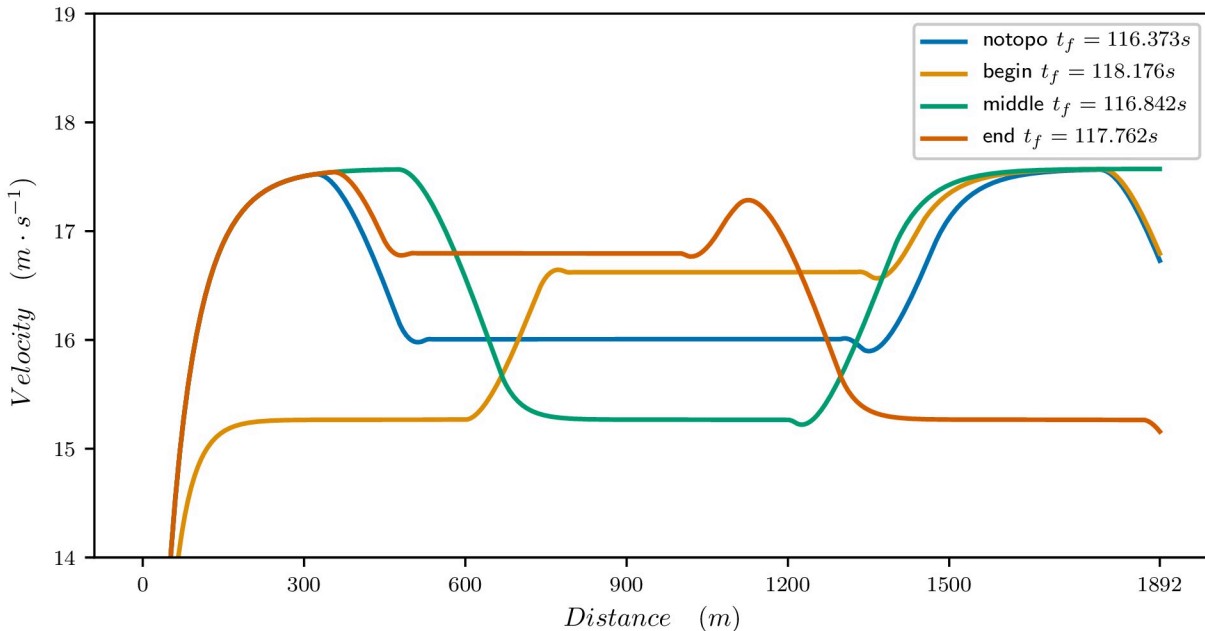

**Fig 13. Effect of bend on the velocity profile for a 1900m race.** Flat straight track (blue), bend for 630m at the beginning of the race (orange), bend for 630m at the middle of the race (green), bend for 630m at the end of the race (red).

for a long time is related to the ability to maintain velocity. The $\dot{V}O2$ starts to decrease when the residual anaerobic energy is too low, and this corresponds to the optimal time to launch the sprint for a long race.

We also understand better the effects of altitude and the bends and find that they are not local effects producing a local perturbation on the velocity strategy, but on the contrary have a global effect on the whole race. Therefore, a good knowledge of the track and training are crucial to adapt the global pacing strategy rather than slowing down because of bends or slopes.

Future works will be devoted to taking additionally into account drafting and the horse psychology [31] since an alternative strategy can be to stay behind to save energy and overtake in the last straight [11].

To maximize an individual horse's potential for winning, it should be entered in races appropriate for its racing ability. Therefore information on a horse speed, endurance or running economy coupled with simulations can help to predict how a horse profile is adapted to some distances to run.

## Acknowledgments

The authors also wish to thank France Galop and Mc Lloyd for providing the data and for their interest in this work. Finally, they are very grateful to Pierre Martinon for his advice on Bocop.

## Author Contributions

**Conceptualization:** Amandine Aftalion.

**Formal analysis:** Amandine Aftalion.

**Funding acquisition:** Amandine Aftalion.

**Investigation:** Quentin Mercier, Amandine Aftalion.

**Methodology:** Amandine Aftalion.

**Project administration:** Amandine Aftalion.

**Software:** Quentin Mercier.

**Supervision:** Amandine Aftalion.

**Validation:** Quentin Mercier, Amandine Aftalion.

**Visualization:** Quentin Mercier, Amandine Aftalion.

**Writing – original draft:** Quentin Mercier, Amandine Aftalion.

**Writing – review & editing:** Quentin Mercier, Amandine Aftalion.

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
