## [Decision Letter · Decision Letter 0]

25 Aug 2020

PONE-D-20-17088

Pacing strategy in horse racing

PLOS ONE

Dear Dr. Aftalion,

Thank you for submitting your manuscript to PLOS ONE. After careful consideration, we feel that it has merit but does not fully meet PLOS ONE’s publication criteria as it currently stands. Therefore, we invite you to submit a revised version of the manuscript that addresses the points raised during the review process.

The abstract should be structured according to the recommendations of PLOS One. It is advisable to include Introduction, Methods, Results (data), and ConclusionDataset is advisable to be available from a proper repository or in Plos One database.

We look forward to receiving your revised manuscript.

Kind regards,

Dalton Müller Pessôa Filho, Ph.D.

Academic Editor

PLOS ONE

Journal Requirements:

Additional Editor Comments:

Please, address all comments and proceeding with the changes in the text and the proper responses in the letter to reviewers.

Reviewers' comments:

Reviewer's Responses to Questions

**Comments to the Author**

1. Is the manuscript technically sound, and do the data support the conclusions?

Reviewer #1: Partly

Reviewer #2: Yes

2. Has the statistical analysis been performed appropriately and rigorously? 

Reviewer #1: N/A

Reviewer #2: Yes

3. Have the authors made all data underlying the findings in their manuscript fully available?

Reviewer #1: Yes

Reviewer #2: Yes

4. Is the manuscript presented in an intelligible fashion and written in standard English?

Reviewer #1: Yes

Reviewer #2: Yes

5. Review Comments to the Author

Reviewer #1: This article addresses a completely new concept in the racehorse. It is particularly interesting to bring a bit of science to a discipline as traditional as gallop racing.

However, perhaps I missed it, but I have the impression that you define the optimal strategy, not from the analysis of a substantial amount of race data, but by having a priori chosen for each distance the profile that seemed optimal to you. It's intellectually embarrassing.

Introduction

Line 4 - « Up to now, only measurements on treadmills have been obtained using masks”. This is true for Thoroughbreds, but track measurements have been made for Standardbred and endurance horses.

Line 14 - Could you define the notion of "pacing strategy"? This notion can be ambiguous in the horse, a species in which there are naturally several paces and where the notion of rhythm or cadence has its own definition in traditional equitation.

Lines 28-29 – “Data are provided by France Galop and are from roughly ten races.”

This is very imprecise. How many races, of what level, how many different horses, of what age, male or female, same day or different day? The physiology of a 2 year old is not the same as that of a 3 or 4 year old. The quality of the terrain varies from one day to the next and influences the fatigue of the horse during the race. Good horses will not necessarily have the same behaviour as poor quality horses ...

Lines 32-34 – Has the accuracy of the GPS system been validated by you? Is it published?

Line 44 – “We use the model developed by [10-12] for the optimal strategy in running and adapt it for horses.” On what scientific basis can you state that the model defined in humans is also suitable for horses? How can you calculate physiological parameters such as strength, anaerobic energy or VO2 without including data obtained from the horse? These are two very different athletes. Can you explain?

Lines 85-86 – “This yields the optimal strategy depending on the length of the race, including the velocity profile and the VO2 profile.” The notion of “optimal strategy” troubles me since you have only analysed 3 races with 3 different horses.

Line 93 – Why did you chose N= one thousand points?

Line 94 – “the points are evenly distributed between s1 and sN.” This means that the distance between the points varies with the running distance, and therefore the accuracy?

Line 127 – “We have chosen three significant races of 1300m, 1900m and 2100m. For each one, we have taken the data of a horse which seems to have run an optimal race.”

What do you mean by significant?

How did you choose this horse? On what criteria? You try to model an optimal race and starting from a race that you judge optimal. It's contradictory. How do you know that this race is optimal?

Why didn't you take data from several horses, over several races of the same distance?

Lines 138-139 – « Nevertheless, a decrease in velocity at the end of such a race takes place even on a flat track.” Is this an observation from your model or a simple statement?

It seems to me that in the horse, the winners are often those who are capable of a final speed peak, or at least capable of maintaining a relatively constant speed while the losers suffocate and lose speed.

Moreover, if we compare the speed curves over 1300 and 1900m, we can see that the speed increases between 1300 and 1900m.

I'm very surprised at the VO2 values calculated in your model. They are relatively equivalent between the 1300 and 2100m races, but much higher for the 1900m race. Were the races and horses chosen really comparable? Doesn't this show that your model, if it is relevant to describe what happens during a race, is still very race-dependent? Can you discuss or propose an explanation?

Figure 7 and comments in the text. You use several terms to describe different conditions (e.g. pink/purple curve, curvature only) to describe the same things.

Line 180 – “The first bend has a strong curvature and therefore reduces drastically the velocity” – How can you say that? you have nearly the same velocities as for the 1900m race

Line 205 –What do you mean by “mean value of the race” ?

Lines 207-224 – I fully understand the idea and what intuitively sounds like an ideal. However, I don't understand how your results lead to this conclusion. That would require a comparison of the races of winners and losers. All you describe is that the best horses have a higher VO2 max, which is nothing new.

Line 232-233. Ref [22] is not really “recent”

Line 240 – Another explanation is that the forces and energy involved on a treadmill are much less than on a track at the same speed.

Table 4. It's very embarrassing that you don't have horses of equal quality or average data on comparable races.

General question: Why didn't you use data from other horses/races to test your assumptions?

Conclusion

Lines 271-272 – I disagree. You need to be more nuanced. You modeled 3 races of 3 different horses over 3 different distances on one track. This allows you to explain what happens and the physiological limits (= everything you develop afterwards). You cannot say that this provides you with the optimal pacing strategy. You would have to analyze data from many races, with horses of different categories.

For me, it would be more explicit to group together figures [3, 5, 8], [4, 6, 9] and [7, 10-12] to facilitate comparisons.

Reviewer #2: I have reviewed "Pacing strategy in horse racing" by Mercier and Aftalion.

Overall, the paper will contribute towards knowledge in the area of the mathematical model able to predict the pacing strategy and worthy of publication. The authors have documented the development of a mathematical model to predict the pacing strategy depending on the distance to run, the shape and topology of the track. A vital issue to address is the discussion of the results mainly concerning VO2max during the 1900 m race. The outcome is clear, but not all conclusions can be justified based on the study's data. I have a few comments and questions that I have detailed below.

Title

The title is clear and informative; however, including the horse breed would be advised since the study was performed in a single breed.

Abstract:

Overall I feel that the research question is not addressed adequately in the abstract. I would like to see a brief rationale as to why the optimal strategy for a horse to run a race is essential in Thoroughbred horses.

The abstract does not contain any data or indication of statistical analysis.

Introduction

Well written and clear. However, the authors do not state a hypothesis.

L5-6: In my opinion, this reference [5] is ancient. You can keep it, but please add a new reference to this information. And authors must specify the interesting breed, the Thoroughbred.

"What is known is that horses have a high aerobic capacity, about twice that of human beings".

This is true for Thoroughbred racehorses. Furthermore, the metabolic demand for Thoroughbred horses during a race is quite different. We have racing distances from 1000 to 3200 m. For example, for 3200 m (2-mile race), the aerobic contribution may be up to 90%.

Please, you must discuss and review the information in the introduction.

Materials and methods

A statement of ethics approval is required before the materials and methods information.

Results

Fig 4., Fig 6. and Fig 9.: "blue is propulsive force." Please note that VO2 (left) is blue too. Improve the caption text to increase readability.

Discussion

L188: "From experiments on human races," What kind of races? Please clarify it.

L236: "The horse of the 1900m race has a very strong VO2max and therefore uses lower anaerobic energy".

Here we must remember that to reach the velocity related to VO2max, and horses need of the anaerobic contribution. Anaerobic contribution (glycolysis) starts to supply ATP from 55% VOmax (see: DOI: 10.1152/japplphysiol.00909.2001). Usually, this velocity/intensity corresponds to the lactate threshold. Besides, Thoroughbred horse locomotor muscles usually contain high percentages of type 2A fibers. Type 2A fibers have a considerable number of capillaries and mitochondria and rely on glycolytic and oxidative metabolism. Also, there are fibers 2AX, a hybrid fibers type. Thus, I wonder how the Thoroughbred does to keep its VO2max during the 1900 m race without the anaerobic metabolism contributing. May be through of the high-energy phosphate system like phosphocreatine pathway? The discussion does not clarify the role of the non-mitochondrial metabolic pathways contribution (i. e., glycolytic, and phosphagen metabolic pathways) during a race. These aspects should be included when discussing and reviewing the results of the current study.

6. PLOS authors have the option to publish the peer review history of their article (what does this mean?). If published, this will include your full peer review and any attached files.

Reviewer #1: No

Reviewer #2: **Yes: **Guilherme Camargo Ferraz

---

## [Author Response · Author response to Decision Letter 0]

23 Sep 2020

Reviewer 1 :

I have the impression that you define the optimal strategy, not from the analysis of a substantial amount of race data, but by having a priori chosen for each distance the profile that seemed optimal to you. It's intellectually embarrassing.

We have had the data for about 30 races of various distances. Because for the time being our model is for a single horse running alone, the model does not take into account interactions between horses : in particular how staying behind for some time saves energy and allows to overtake and win, or the selection of a strategy by the jockey such as starting slowly or on the opposite starting very strongly. Therefore, we have tried to select, among the data we had, horses that ran races close to a race they would have run alone : that is they didn’t seem to have interactions or at least very little, and their strategy didn’t seem to follow from a group strategy or the jockey strategy. Our definition of the optimal race is something close to a race alone. It is not that we have picked the races to match the model, it is just that if a horse runs a race leading it from the beginning to the end, it is easier to fit a model corresponding to a single horse.

Our aim is to extend the model to take into account the two effects of interactions and strategy. But our model as it is cannot predict the time run by the last horse : indeed, the last horse is likely to have run too strongly with respect to his capacity at the beginning of the race and not to be able to run properly till the end. We are working at including strategy and interactions but to start with we needed to have values for the physiological parameters which are consistent for a single horse.

Moreover, if one takes a large amount of races on the same track, on a similar ground, there is not an optimal strategy for all the horses or even for all the winners, because the strategy will depend on interactions: for instance if some horses start too strongly or if everyone starts slowly. In order to predict all types of strategies in the future, we need to fit our one horse model and have the profile for each horse running. Later, we hope to be able to deal with multi horses strategy or change of strategy with respect to the optimal regulation of speed for a single horse.

We have added a paragraph at the beginning of the section « Results » to try and clarify this.

Introduction

Line 4 - « Up to now, only measurements on treadmills have been obtained using masks”. This is true for Thoroughbreds, but track measurements have been made for Standardbred and endurance horses.

We didn’t know this, we have added « for Thoroughbreds » in the text and two references, but we are ready to add other references if the referee is willing to provide some.

Line 14 - Could you define the notion of "pacing strategy"? This notion can be ambiguous in the horse, a species in which there are naturally several paces and where the notion of rhythm or cadence has its own definition in traditional equitation.

We agree with the referee that the notion of pacing is ambiguous, it is used for humans but is probably not as meaningful for horses. We have tried to replace it with « regulating speed to optimize performance », but one of our references has this word in the title, so we could not completely remove the word from our paper.

Lines 28-29 – “Data are provided by France Galop and are from roughly ten races.”

This is very imprecise. How many races, of what level, how many different horses, of what age, male or female, same day or different day? The physiology of a 2 year old is not the same as that of a 3 or 4 year old. The quality of the terrain varies from one day to the next and influences the fatigue of the horse during the race. Good horses will not necessarily have the same behaviour as poor quality horses ...

As pointed out below, when we got the data, only 26 races had been tracked from october 2019 on. Then, there was winter, the covid, and it is only starting slowly again, so there are not many races available. Because it was the end of the season, the races did not involve the best horses. Among these 26 races, France Galop provided us with 10 races with « good » horses.

We have added information on terrain and age. The three races were not the same day. France Galop does not allow us to put the date.

Lines 32-34 – Has the accuracy of the GPS system been validated by you? Is it published?

No, we have not validated the GPS system ourselves. The system was bought by France Galop, the French operational body for flat horse racing in France and they have checked it on about a hundred races comparing the photo every 200m on the last 600m and got accuracy up to the 1000th. More info is on 

https://mclloyd.com/wp-content/uploads/2020/09/HPV2-Data.pdf

The device is now used on every race and some data are available live. We have tried to make our paragraph clearer

Line 44 – “We use the model developed by [10-12] for the optimal strategy in running and adapt it for horses.” On what scientific basis can you state that the model defined in humans is also suitable for horses? How can you calculate physiological parameters such as strength, anaerobic energy or VO2 without including data obtained from the horse? These are two very different athletes. Can you explain?

The previous formulation was very bad. We did not mean that the model defined for humans is suitable for horses. What we meant is that the laws on which the model is based, Newton 2nd law of motion and energy conservation are the same principles that are used for horses. But of course, the model takes into account the physiology of horses. What it does not take into account is the detail of the stride, or on the fact that the horse is going up and down as it gallops. But it did not take into account the detail of the stride for humans either. Nevertheless, this captures the mean velocity on a stride, which is the essential feature leading to velocity computations and interesting information.

The strength of the model is that we do not need to measure all these physiological parameters (strength, VO2, anaerobic energy etc). The values are identified from the GPS data and velocity data and seem to match pretty well what is expected. So of course, we include the data from the horses to compute all the parameters of the problem.

We have tried to improve the explanations in the paper.

Lines 85-86 – “This yields the optimal strategy depending on the length of the race, including the velocity profile and the VO2 profile.” The notion of “optimal strategy” troubles me since you have only analysed 3 races with 3 different horses.

We are not performing a statistical analysis to determine the strategy depending on the distance. Given a horse, a distance to run and a shape of track, the issue is to determine how the horse should regulate its speed in the course of the race to make the best final time. It turns out that for each horse and race analyzed, we have found parameters that match the velocity/gps profile extremely well. If we had analyzed 10 races or 10 horses, it would have been different parameters. But they are always in the same range. It is not the number that is going to provide a global strategy.

But the next step is given the speed profile, to analyze the possible strategy to put when there are two different horses. Each horse and each race corresponds to an optimal strategy. 

Line 93 – Why did you chose N= one thousand points?

Line 94 – “the points are evenly distributed between s1 and sN.” This means that the distance between the points varies with the running distance, and therefore the accuracy?

The discretisation for the optimisation problem is to take one point every two meters, we have added this element. Then for the longest race, 2100 meters, taking N=1000 corresponds to one point every two meters as well. So we could have taken N smaller for shorter races, which we have not done, but it does not raise an accuracy problem.

Line 127 – “We have chosen three significant races of 1300m, 1900m and 2100m. For each one, we have taken the data of a horse which seems to have run an optimal race.”

What do you mean by significant?

How did you choose this horse? On what criteria?

We refer to our paragraph at the very beginning to answer this part and we have rephrased the paragraph in the text.

 You try to model an optimal race and starting from a race that you judge optimal. It's contradictory. 

No, we choose a race that we think optimal and identify all the physiological parameters (VO2, anaerobic energy, tau, f_max) etc on it.

If the horse starts slowly because the jockey wants it like this, then there is no hope that the model will fit this race whatever the parameters.

Nevertheless, in the future, now that we have identified parameters for horses, we hope to be able to model any strategy.

How do you know that this race is optimal?

We assume so by watching the video and checking that it is not full of overtaking, so not too much interaction with other horses. If it was not, the model could not fit any parameter.

Why didn't you take data from several horses, over several races of the same distance?

Because for the moment, very few races have been tracked with the device, and we do not have so many races of the same horse for instance. We also wanted to show the difference on different distances, and not analyze one specific horse.

Lines 138-139 – « Nevertheless, a decrease in velocity at the end of such a race takes place even on a flat track.” Is this an observation from your model or a simple statement?

It is an observation from GPS data : though you have the visual impression that the horse is speeding up, in effect in the last 200m, even the best horses slow down (though slightly of course). This is the case in the data we present, but also in all the data we are aware of.

It seems to me that in the horse, the winners are often those who are capable of a final speed peak, or at least capable of maintaining a relatively constant speed while the losers suffocate and lose speed.

They speed up in the last 600m or so but in the last 200m, they do not maintain this speed peak and in the very end, the best ones are the ones who slow down least. Of course, you cannot see it with speed data every 200m, but you see it clearly with GPS data.

Moreover, if we compare the speed curves over 1300 and 1900m, we can see that the speed increases between 1300 and 1900m.

Not exactly. The maximal speed is higher for the 1300 than the 1900 and the mean speed (distance divided by time) is also higher for the 1300 though the 1900 is run with better horses (3 year old instead of 2).

I'm very surprised at the VO2 values calculated in your model. They are relatively equivalent between the 1300 and 2100m races, but much higher for the 1900m race. Were the races and horses chosen really comparable? 

It is true that the horses were not comparable, now this is stated better, but we do not think that it is a problem for the paper. It shows the effect of the VO2 on the race and that our model is rather powerful.

Doesn't this show that your model, if it is relevant to describe what happens during a race, is still very race-dependent? Can you discuss or propose an explanation?

Of course, the model is race and horse dependent and hopefully because not all races are run the same. For instance, the 1900m is run in 116 seconds while the 2100 in 131, so it is obvious from the total time that the horses of the 2100 were not as strong and the interest of our model is that it determines the physiological parameters which are different.

Figure 7 and comments in the text. You use several terms to describe different conditions (e.g. pink/purple curve, curvature only) to describe the same things.

Everything is now pink

Line 180 – “The first bend has a strong curvature and therefore reduces drastically the velocity” – How can you say that? you have nearly the same velocities as for the 1900m race

No, maybe the scale is not easy to read but the data stay below 17 (the identification slightly above), for the 2100, while for the 1900, they top speed at 17.6 at the beginning of the race, and there is a difference in the acceleration profile. Nevertheless, we have removed « drastically ».

Line 205 –What do you mean by “mean value of the race” ?

We have changed the wording, we mean that the VO2 reaches a plateau, which is not the maximal value of the race, which is reached before.

Lines 207-224 – I fully understand the idea and what intuitively sounds like an ideal. However, I don't understand how your results lead to this conclusion. That would require a comparison of the races of winners and losers. All you describe is that the best horses have a higher VO2 max, which is nothing new.

Actually, we can vary slightly the physiological parameters in the model to see the evolution of the optimal velocity. Once we have identified parameters on a horse, we can thus simulate a stronger departure or other situations and see how the velocity evolves. We do not need extra data. This is what we comment. We do not show figures because there are already many figures in the paper. We include a figure in the pdf file for the answer for the 1300m, where the blue is the identified solution, the orange has a slow start (and therefore speeds up more at the end but in total is 4 hundredth slower ) and the green has a fast start, slows down more at the end and is also about 4 hundredth slower)

Line 232-233. Ref [22] is not really “recent”

We have removed the word recent.

Line 240 – Another explanation is that the forces and energy involved on a treadmill are much less than on a track at the same speed.

We have added forces, we are not sure of any treadmill at the same speed as the ones in the race we present.

Table 4. It's very embarrassing that you don't have horses of equal quality or average data on comparable races.

It is the very beginning of the tracking system in France. In 2019, only 26 races were officially tracked as we mentionned above and very little has been done since. It is starting again now.

Nevertheless, in our point of view, it would be a different work to compare a same horse on different distances. As for average data, it is not the same type of mathematics since it would be statistics, whereas we need data for a single horse on a single race for identification.

Our model, as it is, once we have identified on a race, we can predict the velocity for the same horse on another distance or with another topography.

General question: Why didn't you use data from other horses/races to test your assumptions?

As explained before, this was the very beginning of the tracking so that not so many data were available and we thought it was interesting to show the first results we got on these simulations.

Conclusion

Lines 271-272 – I disagree. You need to be more nuanced. You modeled 3 races of 3 different horses over 3 different distances on one track. This allows you to explain what happens and the physiological limits (= everything you develop afterwards). You cannot say that this provides you with the optimal pacing strategy. You would have to analyze data from many races, with horses of different categories.

We have changed the sentence accordingly.

For me, it would be more explicit to group together figures [3, 5, 8], [4, 6, 9] and [7, 10-12] to facilitate comparisons.

We understand that this would have advantages but if one wants to see the relationships between speed, force and VO2, it is better to have them grouped together for the same horse, so we prefer to keep our initial grouping of figures.

Reviewer 2 :

 A vital issue to address is the discussion of the results mainly concerning VO2max during the 1900 m race. The outcome is clear, but not all conclusions can be justified based on the study's data. 

Title

The title is clear and informative; however, including the horse breed would be advised since the study was performed in a single breed.

Thoroughbred has been added to the title

Abstract:

Overall I feel that the research question is not addressed adequately in the abstract. I would like to see a brief rationale as to why the optimal strategy for a horse to run a race is essential in Thoroughbred horses.

The abstract does not contain any data or indication of statistical analysis.

We have rewritten the asbtract and tried to take into account the referee’s comments about the research question. There maybe a misunderstanding about the maths we are using : we definitely do not do any statistics, what we use are deterministics mathematics, that is coupled ordinary differential equations. We have tried to clarify this.

Introduction

Well written and clear. However, the authors do not state a hypothesis.

Since we use a deterministic model of coupled ordinary differential equations leading to an optimal control problem, we perform simulations with adequate parameters and analyze the results of the simulations, but with these types of mathematics, we cannot make hypothesis, we just follow the simulations results. The only hypothesis that we can think of is that some horses run an « optimal » race and therefore there is a hope to fit the data with our model. We have added a paragraph related to this issue.

L5-6: In my opinion, this reference [5] is ancient. You can keep it, but please add a new reference to this information.

We have added a more recent reference, which is a book but if the referee has others to suggest, we are willing to add them.

 And authors must specify the interesting breed, the Thoroughbred.

Done

"What is known is that horses have a high aerobic capacity, about twice that of human beings".

This is true for Thoroughbred racehorses. Furthermore, the metabolic demand for Thoroughbred horses during a race is quite different. We have racing distances from 1000 to 3200 m. For example, for 3200 m (2-mile race), the aerobic contribution may be up to 90%.

Please, you must discuss and review the information in the introduction.

We have added a reference and discussed this issue but it seems that it is not so well known in the literature. The relationship that exists between performance and anaerobic capacity remains to be determined and only estimates according to distances exist in the literature.

Materials and methods

A statement of ethics approval is required before the materials and methods information.

We have not performed any experiments on horses. The races are official French races operated by France Galop, the French governing body for French flat races. For these races, they have added a GPS device of 90g on the saddle. So the only effect is to slightly increase the weight by 90g. France Galop has signed an agreement about horse welfare in March 2016 at the annual Agricultural Fair in Paris. It can be found on

http://www.france-galop.com/en/our-responsibilities

but i do not see how we could write of statement of ethics, since our only task was to receive position and velocity data. In reference 11 where they received data from English races, there is not ethics statement.

Results

Fig 4., Fig 6. and Fig 9.: "blue is propulsive force." Please note that VO2 (left) is blue too. Improve the caption text to increase readability.

We have changed the color of VO2 to green.

Discussion

L188: "From experiments on human races," What kind of races? Please clarify it.

from 400m to 1500m (that is of similar duration of the races we analyze here), added to the text

L236: "The horse of the 1900m race has a very strong VO2max and therefore uses lower anaerobic energy".

Here we must remember that to reach the velocity related to VO2max, and horses need of the anaerobic contribution. Anaerobic contribution (glycolysis) starts to supply ATP from 55% VOmax (see: DOI: 10.1152/japplphysiol.00909.2001). Usually, this velocity/intensity corresponds to the lactate threshold. Besides, Thoroughbred horse locomotor muscles usually contain high percentages of type 2A fibers. Type 2A fibers have a considerable number of capillaries and mitochondria and rely on glycolytic and oxidative metabolism. Also, there are fibers 2AX, a hybrid fibers type. Thus, I wonder how the Thoroughbred does to keep its VO2max during the 1900 m race without the anaerobic metabolism contributing. May be through of the high-energy phosphate system like phosphocreatine pathway? The discussion does not clarify the role of the non-mitochondrial metabolic pathways contribution (i. e., glycolytic, and phosphagen metabolic pathways) during a race. These aspects should be included when discussing and reviewing the results of the current study.

We have tried to clarify these aspects in the text and we have added a figure which plots the anaerobic energy vs the distance to the finish line. From this, it is clear that in the middle of the race where the aerobic consumption is maximum, the horse of the 1900m uses less anaerobic energy than the others, since the slope is smaller. As explained in ref. 10, when the VO2max is higher, then less anaerobic energy is needed to maintain the same speed. So the anaerobic metabolism contributes for the 1900m horse, but less than the others. Our model treats the anaerobic energy as a global tank so that we are unable to be more precise about the pathway. 

If we take into account the running economy (VO2max/speed), then the horse of the 1900m has a much higher RE than the others.

---

## [Decision Letter · Decision Letter 1]

19 Oct 2020

Optimal speed in Thoroughbred horse racing

PONE-D-20-17088R1

Dear Dr. Aftalion,

We’re pleased to inform you that your manuscript has been judged scientifically suitable for publication and will be formally accepted for publication once it meets all outstanding technical requirements.

Kind regards,

Dalton Müller Pessôa Filho, Ph.D.

Academic Editor

PLOS ONE

Additional Editor Comments (optional):

Reviewers are agreement about the considerable changes in the manuscript and considered it suitable to be published in Plos One.

Reviewers' comments:

Reviewer's Responses to Questions

**Comments to the Author**

1. If the authors have adequately addressed your comments raised in a previous round of review and you feel that this manuscript is now acceptable for publication, you may indicate that here to bypass the “Comments to the Author” section, enter your conflict of interest statement in the “Confidential to Editor” section, and submit your "Accept" recommendation.

Reviewer #1: All comments have been addressed

Reviewer #2: All comments have been addressed

2. Is the manuscript technically sound, and do the data support the conclusions?

Reviewer #1: Yes

Reviewer #2: Yes

3. Has the statistical analysis been performed appropriately and rigorously? 

Reviewer #1: Yes

Reviewer #2: N/A

4. Have the authors made all data underlying the findings in their manuscript fully available?

Reviewer #1: Yes

Reviewer #2: Yes

5. Is the manuscript presented in an intelligible fashion and written in standard English?

Reviewer #1: Yes

Reviewer #2: Yes

6. Review Comments to the Author

Reviewer #1: Thank you for considering each of my remarks, responding to them in detail and modifying your manuscript accordingly.

The value of your model and the fact that it needs to be adapted to each situation may not yet be sufficiently reflected in the text. Perhaps more emphasis should be placed on the fact that this is a preliminary study, and that not only the racing distance varies, but also the age of the horses, their level of fitness and probably their aptitude, the racing conditions ...

Nevertheless, this study is original, and the analyses well carried out. It deserves to be published.

Reviewer #2: The reviewer would like to thank the authors for taking the time to provide additional analysis and discussion of their research data and for making extensive additions to the manuscript to address comments from the initial review. The authors have addressed all recommendations for revision, and therefore the reviewer recommends accepting the manuscript for publication.

7. PLOS authors have the option to publish the peer review history of their article (what does this mean?). If published, this will include your full peer review and any attached files.

Reviewer #1: No

Reviewer #2: **Yes: **Guilherme C Ferraz

---

## [Editor Report · Acceptance letter]

30 Oct 2020

PONE-D-20-17088R1 

Optimal speed in Thoroughbred horse racing 

Dear Dr. Aftalion:

I'm pleased to inform you that your manuscript has been deemed suitable for publication in PLOS ONE. Congratulations! Your manuscript is now with our production department. 

Kind regards, 

on behalf of

Prof. Dr. Dalton Müller Pessôa Filho 

Academic Editor

PLOS ONE